# Intratumoral Microbiome: Impact on Cancer Progression and Cellular Immunotherapy

**DOI:** 10.3390/cancers18010100

**Published:** 2025-12-29

**Authors:** Georgy Leonov, Antonina Starodubova, Oleg Makhnach, Dmitry Goldshtein, Diana Salikhova

**Affiliations:** 1Department of Cardiovascular Pathology and Diet Therapy, Federal Research Center of Nutrition, Biotechnology and Food Safety, Moscow 109240, Russia; avs.ion@yandex.ru; 2Stem Cell Genetics Laboratory, Research Centre for Medical Genetics, Moscow 115522, Russia; buben6@yandex.ru (O.M.); goldshteyn_dv@pfur.ru (D.G.); diana_salikhova@bk.ru (D.S.); 3Therapy Faculty, Pirogov Russian National Research Medical University, Moscow 117997, Russia; 4Institute of Molecular and Cellular Medicine, RUDN University, Moscow 117198, Russia

**Keywords:** intratumoral microbiome, tumor microenvironment, immune response, cell therapy, gut-tumor axis, antitumor immunity, CAR-T cells

## Abstract

Cellular immunotherapy is a growing field that has shown significant success in treating oncological diseases, utilizing living immune cells such as CAR-T cells and NK cells. However, challenges remain, particularly low efficacy in solid tumors and immunosuppression within the tumor microenvironment. Recent research supports the long-standing hypothesis that organs traditionally viewed as sterile, including tumor tissues, harbor diverse microbial communities, referred to as the intratumoral microbiome. This microbiome is recognized as an important element in cancer development and progression, exhibiting both stimulatory and inhibitory effects. The intratumoral microbiota strongly influences immune cell activity and regulates local and systemic immune responses. The aim of this review is to summarize data on the role of the intratumoral microbiome, with a primary focus on its bacterial component, in the development and progression of cancer, as well as the interaction of microorganisms in tumor tissue with immune cells, especially in the context of cellular immunotherapy.

## 1. Introduction

Immunotherapy represents a rapidly advancing field that has demonstrated significant progress in recent years with regard to the treatment of oncological diseases. The findings of research conducted to date indicate that this therapeutic method is effective in the suppression of tumor growth and the improvement in clinical outcomes. The treatment approach has been shown to stimulate the host’s immune system, thereby restoring immunological functions and eliminating tumor cells over an extended period of time [1]. The employment of immunotherapy can be considered as a standalone treatment strategy or in combination with traditional methods, thereby demonstrating the potential for a synergistic effect [2]. Cellular immunotherapy refers to the therapeutic administration of living immune cells to treat disease, particularly malignancies. In contrast to vaccines, which are defined as prophylactic interventions intended to prime or boost immune responses against infectious agents, cellular immunotherapy is applied as a treatment strategy [3]. It may be classified as active immunotherapy, exemplified by dendritic cell-based immunotherapy aimed at inducing or enhancing antitumor immune responses, or as passive immunotherapy, such as adoptive cell transfer (ACT), in which autologous or allogeneic lymphocytes with inherent antitumor activity are administered. When immune cells are used in their unmanipulated form and delivered via infusion or in situ application, this approach is more appropriately defined as cell therapy, whereas ex vivo-manipulated or functionally enhanced cells fall within the scope of advanced cellular immunotherapy [4]. Non-specific oncology cellular immunotherapy (OCI) encompasses a variety of cell types, including dendritic cells (DCs), natural killer cells (NK), cytokine-induced killer cells (CIKs), tumor-infiltrating lymphocytes (TILs), lymphocyte-activated killer cells (LAKs), and killer-induced macrophages (MAKs), among others. In specific OCI, immune cell activation is mediated by tumor antigens and specific stimulating factors. These methods include TIL-based therapy, T-cell receptor-transfer (TCR-T) technologies, and chimeric antigen receptor-positive T-cell (CAR-T) immunotherapy [5,6,7,8,9,10,11]. The U.S. Food and Drug Administration (FDA) has granted regulatory approval to several cell-based medicinal products. In the domain of immune cell therapies, a number of CAR-T products have received approval for the treatment of hematologic malignancies, including certain forms of leukemia, lymphoma, and multiple myeloma. Examples include tisagenlecleucel (Kymriah, Basel, Switzerland), axicabtagene ciloleucel (Yescarta, El Segundo, CA, USA), and idecabtagene vicleucel (Abecma, Lawrence, NJ, USA) [12]. Furthermore, the TIL drug lifileucel (Amtagvi, San Carlos, CA, USA) has been approved for the treatment of adult patients with unresectable or metastatic melanoma [13]. However, immunotherapy still faces many challenges. These include relatively low efficacy in solid tumors, in part due to heterogeneity and the difficulty of targeting specific antigens. Other challenges include tumor microenvironment immunosuppression and complications in cellular drug delivery [14]. Furthermore, there are also safety concerns [15]. Immunotherapy has been shown to be associated with a risk of adverse effects, including unfavorable outcomes [16,17]. The assessment of biosafety in cell therapy is based on a risk-oriented, multi-parameter evaluation that translates major hazards into operational principles. These principles include toxicity, oncogenicity, tumorigenicity, teratogenicity, immunogenicity, biodistribution, and cell-product quality. These principles are tested in a layered way (in vitro, in vivo, and clinically) to support an overall risk–benefit evaluation [18]. Nevertheless, the development of next-generation drugs and additional approaches aimed at increasing the effectiveness and safety of cell therapy is ongoing [19]. Immune system function and antitumor immune responses are closely associated with the activity of microorganisms that colonize the human body [20].

The human microbiome both positively and negatively contributes to carcinogenesis by influencing signaling pathways involved in inflammation, DNA repair, and stability [21]. Moreover, the gut microbiome is widely recognized as a factor that significantly influences the regulation of local and systemic immune responses in mouse models and human studies [22]. It has been demonstrated that the composition of the microbiota, particularly the levels of abundance of individual bacterial taxa, exerts a substantial influence on the efficacy of the host’s immune response [23]. The relationship between the microbiome and immune cells used in immunotherapy is a relatively new and understudied area of research [24]. A substantial body of research has demonstrated the correlation between specific taxa of microbiota and the manifestation of a response to immunotherapy. However, the comparability of these findings is constrained by the paucity of shared species among studies. Clinical data underscores the deleterious effect of antibiotics administered prior on treatment outcomes and survival. Broad-spectrum antibiotics reduce microbial diversity, disrupt the balance between beneficial and unfavorable taxa, promote the growth of immunosuppressive fungi, and alter the intestinal mucosa, which disrupts host–microbiome interactions and modulate the immune response [25].

The long-standing hypothesis that microorganisms may inhabit tumor tissues of organs outside the gastrointestinal tract has only recently gained support through advanced technological methodologies, revealing that organs and tissues traditionally considered sterile can in fact harbor diverse microbial communities [26]. A range of molecular biology techniques are employed in the study of the intratumoral microbiome, including next-generation sequencing (NGS) methods such as the sequencing of the 16S rRNA gene and whole-genome shotgun sequencing (WGS). To validate NGS results and confirm the presence, localization, and metabolic activity of bacteria directly within the tumor, methods such as fluorescence in situ hybridization (FISH) and fluorescently labeled amino acid incorporation technology, including 3D imaging (miCDaL), are performed [27,28].

The handling of tumor tissues presents significant challenges due to the low concentration of bacterial DNA (low biomass), which makes samples highly susceptible to contamination from reagents, the laboratory environment, and adjacent tissues [29]. Additionally, the non-culturable nature of some types of microorganisms and the heterogeneity of the tumor microenvironment are significant aspects [30]. Moreover, the analysis of bioinformatics data presents a considerable challenge, given the possibility of false-positive results in the identification of bacterial reads [31]. A substantial body of evidence suggests a close association between a number of malignant neoplasms and infection by microorganisms, including bacteria, viruses, and fungi. Intratumoral microbiota is regarded as an important element in the development of cancer, exhibiting both stimulatory and inhibitory effects on tumor progression [32,33].

The aim of this review is to summarize data on the role of the intratumoral microbiome, with a primary focus on its bacterial component, in the development and progression of cancer, as well as the interaction of microorganisms in tumor tissue with immune cells, especially in the context of cellular immunotherapy. For this narrative review, a comprehensive literature search was conducted using multiple scholarly databases and indexing platforms, including Scopus, Web of Science (WoS), PubMed/MEDLINE, and Google Scholar. These databases were queried using relevant keywords and search terms aligned with the main concepts of the review to identify pertinent peer-reviewed publications.

## 2. Origin of the Intratumoral Microbiota and Relationships with Other Human Microbiomes

The human body contains a multitude of ecological niches that are populated by microorganisms. Furthermore, these distinct microbiomes have the capacity to influence each other [34]. For instance, the oral cavity is in direct connection with the gut, and members of the oral microbiome can enter the gut via the enteral route [35]. While the oral–intestinal barrier is generally considered to be effective in preventing the translocation of most microorganisms, there are instances in which these first lines of defense are compromised, allowing for oral–intestinal transfer. Low gastric acidity, particularly due to the use of proton pump inhibitors, has been demonstrated to shift the composition of the gut microbiota toward oral profiles [36]. A recent study showed that *F. nucleatum* resists gastric acid due to the presence of erucic acid in the cell membrane [37]. In addition, intestinal bacteria themselves may be a potential barrier to the growth of oral microorganisms in the intestine, providing colonization resistance [38]. However, some studies have not found significant colonization of the gut microbiome by oral bacteria in either healthy individuals or patients taking long-term antibiotics [39,40]. The hematogenous route of bacterial transmission from the oral cavity to the intestine involves oral bacteria, particularly pathobionts entering the bloodstream during transient bacteremia induced by daily activities such as toothbrushing or chewing, as well as by dental procedures that disrupt periodontal barriers [41]. These microbes survive in circulation by resisting immune clearance and interacting with endothelial cells, then translocate to intestinal vascular beds, where inflammation enhances adhesion and mucosal seeding, contributing to diseases like colorectal cancer (CRC) and inflammatory bowel disease [42]. Other microbiomes, such as the skin or genital microbiome, can interact with the gut and oral microbiomes primarily through various metabolites [43,44]. Gut-derived metabolites, such as short-chain fatty acids (SCFAs), secondary bile acids, and tryptophan catabolites, reach the skin via the circulatory system. There, they modulate cutaneous barrier function, local immune responses, and the composition and activity of resident microbial communities. This shapes susceptibility to inflammatory dermatoses and ageing-related dysbiosis [45]. A meta-analysis found that, compared with low fiber intake, high fiber intake was consistently associated with an improved therapeutic response to cancer immunotherapy (pooled odds ratio: 5.79). In experimental models, restricting methionine and cysteine, as well as reducing leucine and glutamine intake, slowed tumor progression. Meanwhile, a combination of checkpoint inhibitors with intermittent fasting or a fasting-mimicking diet significantly reduced tumor volume in mice bearing melanoma. In patients, higher concentrations of SCFAs and lactic acid-producing bacteria (*Faecalibacterium prausnitzii* and *Akkermansia muciniphila*) correlated with increased objective response rates [46]. In an in vivo study, it has been demonstrated that skin damage can result in alterations to the intestinal microbiome, particularly through the increased expression of *Reg3* and *Muc2* in intestinal epithelial cells [47]. Through the gut–lung axis, the same classes of metabolites influence pulmonary immunity (for example by altering alveolar macrophage and regulatory T-cell function), which in turn modifies the ecological niche in the respiratory tract and can drive shifts in lung microbial composition and infection susceptibility [48].

The origins of intratumoral microbiota remain a subject of investigation. Intratumoral bacteria can arrive at a tumor site via several non-exclusive routes, including local overgrowth and invasion from adjacent mucosa, blood- and lymph-mediated dissemination, migration along anatomical connections, and transport within circulating tumor cells [49]. Local invasion from adjacent tissue is a primary route, particularly in gastrointestinal cancers where epithelial barriers are compromised. In colorectal cancer, bacteria from the gut lumen become enriched at tumor sites, while in gastric cancer, *Helicobacter pylori* is known to invade the gastric epithelium directly [50]. Thus, it has been demonstrated that microorganisms capable of penetrating the intestinal mucosal barrier and entering the bloodstream can disseminate to other organs within the gastrointestinal tract, thereby contributing to tumor development [50]. Bacteria can also migrate along direct anatomical connections between organs. For instance, in pancreatic ductal adenocarcinoma (PDAC), gut bacteria can travel from the upper gastrointestinal tract into the pancreatic tumor. Similarly, microbes can translocate from the gut to the liver via the portal circulation, where they have been implicated in the development of hepatocellular carcinoma. [51]. Furthermore, the presence of intestinal bacteria has been shown to promote the dissemination of colorectal cancer metastases to the liver by enhancing the formation of a premetastatic niche and attracting metastatic cells [52]. The study showed that *Fusobacterium* is preferentially associated with cancer cells in metastases, and treatment with metronidazole reduces bacterial load, cell proliferation and tumor growth [53]. Another study demonstrated that in patients with colon cancer, compared with the control groups and patients with adenomatous polyps, there was an enrichment of oral biofilm bacteria (*Fusobacterium*, *Gemella*, *Parvimonas*, *Granulicatella*, *Leptotrichia*, *Peptostreptococcus*, *Campylobacter*, *Selenomonas*, *Porphyromonas, Prevotella*) and a number of intestinal taxa (*Phascolarctobacterium*, *Bacteroides*, *Tyzzerella*, *Desulfovibrio*, *Eubacterium*, *Lachnospiraceae*), with the species *F. nucleatum* dominating [54]. It was shown that alpha diversity in tumor samples was lower than in healthy tissue, in addition to a decrease in the representation of *Bacteroides*, *Lachnospiraceae*, *Clostridiales* and *Clostridium* and an increase in *Enterococcus*, *Streptococcus* in tumors [55]. The *Bacilli* class showed a bidirectional positive association with malignant melanoma. The *Betaproteobacteria* and *Gammaproteobacteria* classes demonstrated a causal relationship with an increased risk of developing malignant melanoma and basal cell carcinoma, respectively. In a reverse Mendelian analysis, malignant melanoma was associated with reduced abundance of members of the *Bacteroidetes* (*Bacteroidota*) phylum [56]. In the lung cancer model, *Veillonella parvula*-induced lower respiratory tract dysbiosis resulted in decreased survival, increased tumor burden, and the development of an inflammatory phenotype mediated by the cytokine IL-17 [57]. Experimental data have shown that intestinal bacteria, including *Enterococcus faecalis* and *Escherichia coli*, are able to migrate to the pancreas, forming an intrapancreatic microbiome primarily represented by the phyla *Proteobacteria (Pseudomonadota)*, *Bacteroidetes (Bacteroidota*), and *Firmicutes* (*Bacillota*). Mice with oncogenic pancreatic mutations showed a progressive enrichment of *Bifidobacterium pseudolongum* and *Actinobacteria*, indicating the involvement of intestinal dysbiosis in the development and progression of pancreatic adenocarcinoma [58].

## 3. The Role of the Intratumoral Microbiome in Cancer Progression

### 3.1. Ecology and Composition of the Intratumoral Microbiome

The intratumoral microbiome is constituted by a consortium of microorganisms, comprising bacteria, fungi, and viruses, that inhabit tumor tissue [59]. The potential mechanisms of microbes’ entry into the tumor include damaged mucosal membranes, direct migration from adjacent normal tissue, and hematogenous dissemination [60]. It is noteworthy that bacteria are predominantly localized inside cells [61]. These bacteria actively invade host cells via zipper or trigger mechanisms, often exploiting actin polymerization and depolymerization to induce membrane protrusions that engulf them into a vacuole [62,63]. Target cells include epithelial, endothelial, and keratinocytes, as well as various immune cells like macrophages [64,65]. After entering host cells via trigger or zipper mechanisms, bacteria end up in phagosomes, which normally fuse with lysosomes to destroy the invaders [66]. However, bacteria have evolved multiple survival strategies: *Mycobacterium tuberculosis* uses an impenetrable envelope, detoxification, and Rab-5A protein to block lysosomal fusion. *Listeria*, *Rickettsia*, and *Shigella* escape into the cytosol by hijacking the host cytoskeleton via actin polymerization. *Rickettsia*, *Burkholderia*, *Listeria* monocytogenes, and *Shigella flexneri* exploit actin to spread between cells, forming double-membrane vacuoles in new host cells. Some bacteria also inhibit autophagy to evade destruction [67]. In various cancers, specific bacteria have been detected in tumor tissues and associated with particular cell types. In oral squamous cell carcinoma (OSCC), *Fusobacterium* and *Treponema* have been linked to macrophages and aneuploid epithelial cells [68]. *Helicobacter pylori* is found in gastric cancer tissue, with evidence indicating its ability to attach to and potentially invade gastric epithelial cells [69]. The accumulation of microorganisms in tumor tissue is multifactorial. Tumors have been observed to create hypoxic, permeable, and immunosuppressive niches that facilitate selective persistence of intratumoral microbes rather than efficient immune clearance. Consequently, intratumoral microbes signal through innate immune pathways that bias myeloid populations toward suppressive phenotypes and promote T-cell dysfunction/exclusion. Some bacteria have been observed to directly inhibit antitumor lymphocytes [70,71]. The process of hematogenous transport is facilitated by increased vascularization and vascular permeability of tumor tissue. Tumor tissue is characterized by a high nutrient content and the presence of specific metabolites (e.g., ribose, aspartic acid, etc.), which attract bacteria [72]. Furthermore, the distribution of microorganisms within the tumor is found to be nonuniform, with their habitation occurring within highly organized microniches [68].

The tumor microbiome demonstrates low biomass. However, this parameter is influenced by various factors, including cancer type, microenvironmental conditions, and the implementation of antibiotic treatment [73,74]. The bacterial composition of tumors in various organs has been demonstrated to have distinct characteristics. The composition of the microbiome is also a spatial and ecological property. In colorectal cancer, proximal tumors frequently exhibit polymicrobial biofilms (often extending into adjacent mucosa) [75]. The composition of the taxa *Alistipes*, *Blautia*, *Pasteurellales*, and *Porphyromonas* was correlated with the clinical characteristics of patients with gastrointestinal cancer, particularly colorectal cancer [76]. In lung cancer, compositional shifts have been linked to clinical phenotypes and exposures. Spatial metatranscriptomic profiling indicates that bacterial burden can form a gradient from normal tissue and tertiary lymphoid structures towards tumor cells and airways. This correlates with oncogenic pathways, such as β-catenin [77]. Studies have demonstrated that the composition of the lung microbiota undergoes significant alterations in lung cancer. An increased abundance of *Thermus* has been observed in patients with advanced-stage tumors, whereas *Legionella* is more prevalent in those with metastatic disease [78]. Moreover, metabolic pathways linked to smoking are enriched in the lung microbiota of cancer patients, with specific taxa such as *Acidovorax* showing notable associations with these changes [79]. The study revealed that the predominant taxa at the phylum level in esophageal carcinoma were *Bacteroidetes*, *Firmicutes*, *Proteobacteria*, *Fusobacteria*, and *Actinobacteria*. A substantial increase in the abundance of *Firmicutes* was observed, accompanied by a concurrent decrease in *Proteobacteria* [80]. Increased abundance of intratumoral *Neisseriaceae* has been demonstrated to promote accelerated lymph node metastasis in squamous cell carcinoma of the oropharyngeal tonsils [81]. Furthermore, the presence of certain tumor microorganisms, such as *Brevundimonas* and *Staphylococcus*, has been identified as a contributing factor to the development of distant metastases in breast cancer cases [82]. *F. nucleatum*, a bacterium detected in both primary tumors and metastatic lesions across various cancer types, is of particular interest, as it substantiates its potential role in the maintenance and spread of tumorigenesis [74].

### 3.2. Pro-Tumor Mechanisms of the Intratumoral Microbiome

The intratumoral microbiome exerts multifaceted effects on the processes of carcinogenesis. Extensive research has explored the mechanisms through which microbial communities contribute to tumor initiation and progression. To date, four principal mechanisms have been identified: induction of DNA damage with a consequent increase in mutation rates, modulation of tumor-associated signaling pathways, alteration of host immune responses, and the initiation or amplification of inflammatory processes [83]. The process of DNA mutation induction serves as a primary, key mechanism in the context of microbial carcinogenicity [84]. Multiple studies indicate that tumor viruses promote oncogenesis through diverse direct and indirect mechanisms. Notably, hepatitis B virus (HBV) and human papillomavirus (HPV) can integrate their genomes into host chromosomes, driving dysregulation of cellular processes, uncontrolled proliferation, and malignant transformation. In parallel, certain carcinogenic bacteria can induce host DNA damage via multiple pathways, thereby fostering genetic alterations that facilitate tumorigenesis. Some bacteria have been shown to cause DNA damage and mutations [60]. *E. coli* that expresses the genotoxin colibactin (PKS locus) has been shown to induce double-strand DNA breaks and promote the development of colorectal cancer. It has been demonstrated that other pathogenic *E. coli* strains are capable of injecting the genotoxin UshA via the type III secretion system, thereby inducing DNA damage in intestinal epithelial cells [85]. *Bacteroides fragilis*-produced toxin (Bft) has been reported to increase levels of reactive oxygen species (ROS) in intestinal epithelial cells, leading to oxidation and DNA damage to host tissues and, consequently, malignant transformation of cells [86].

Following initiation, the modulation of tumor-associated signaling pathways serves as a critical secondary mechanism for the development and progression of cancer. The signaling pathways that facilitate cell movement, growth, survival, and metabolism are critical to the fundamental functioning of normal and tumor tissues. The intratumoral microbiome may play a role in tumor-associated signaling cascades. The modulation of various signaling pathways by microbes is a critical factor in the development and progression of cancer [87]. Representatives of the intratumoral microbiome have been shown to modulate cascades of signaling pathways such as WNT/β-catenin, NF-κB, Toll-like receptors (TLR), ERK, PI3K, and RhoAROCK in tumor cells, thereby influencing carcinogenesis [83].

The Wnt/β-catenin signaling pathway is a conserved signaling pathway that regulates embryonic development and tissue homeostasis. β-catenin has been observed to bind to the TCF transcription complex and translocate to the nucleus, where it has been shown to stimulate transcription of downstream target genes such as c-Myc, cyclin D1, MMP, or survivin. This process contributes to the development of various types of cancer. [88]. It has been demonstrated that several intratumoral microorganisms possess the capacity to activate the Wnt/β-catenin signaling pathway. Annexin A1, a regulatory protein that has been recently implicated in this pathway, has been shown to enhance the expression of FadA, a secreted protein produced by *F. nucleatum* in colorectal tumors, via E-cadherin. [89]. The study demonstrated that bacteria belonging to the genus *Veillonella* upregulate the ERK and PI3K signaling pathways in lung cancer. Activation of the PI3K pathway promotes cancer cell proliferation and is considered an early event in lung tumorigenesis [90]. Moreover, stimulation of Toll-like receptor 4 (TLR4) by bacterial lipopolysaccharides induces the expression of proinflammatory genes, contributing to the progression of tumor-promoting inflammation across various cancer types [91]. The NF-κB signaling pathway plays a pivotal role in the development of chronic microbial-induced inflammation. It has been demonstrated that microorganisms can trigger the NF-κB pathway by activating the β-catenin and TLR signaling cascades. This activation results in the release of proinflammatory mediators and the development of a persistent inflammatory state. This process is bidirectional: immune cells recruited to the site of inflammation release cytokines and chemokines, which in turn increase NF-κB activation, creating a positive feedback loop that promotes tumor development and progression [92]. A recent study demonstrated that *Faecalibacterium prausnitzii* and the intestinal metabolite tyrosol exert antitumor effects by inhibiting the activation of the HIF-1α/NF-κB signaling pathway, resulting in reduced levels of reactive oxygen species and inflammatory factors [93]. The presence of bacteria within tumor cells has been shown to suppress the RhoA/ROCK signaling pathway, thereby facilitating the adaptation of circulating tumor cells to fluid shear stress through cytoskeletal remodeling and enhancing their potential for distant colonization [94]. Elevated concentrations of *Staphylococcus*, *Lactobacillus*, and *Streptococcus* have been detected within breast cancer cells, where these bacteria inhibit the RhoA/ROCK pathway, a key regulator of cytoskeletal organization. This inhibition enables tumor cells to withstand mechanical stress encountered in the circulatory system, reducing cellular damage and promoting metastatic dissemination [95]. However, it has been demonstrated that certain microorganisms can exert a beneficial influence by impeding the progression of cancer. Bacterial peptides are presented on tumor and antigen-presenting cell HLA molecules, creating neo-like targets that can be recognized by T cells and potentially broaden antitumor repertoires. Intracellular tumor bacteria and their microbe-associated molecular patterns can activate pattern-recognition pathways, interfacing with innate sensing hubs that shape T cell priming in the tumor microenvironment [96]. Microbial-derived metabolites (e.g., short-chain fatty acids (SCFAs) can reprogram local immunity and myeloid states; colorectal models show that leveraging microbiota metabolism can increase intratumoral SCFAs and augment chemotherapy and immune activation [97]. Therefore, *Lactobacillus* casei and *Lactobacillus reuteri* have been documented to impede the proliferation and migration of pancreatic cancer cells by attenuating TLR4 signaling. Furthermore, these bacterial species counteract the induction of the M2 macrophage phenotype by pancreatic cancer cells and promote the differentiation toward the pro-inflammatory M1 macrophage subtype [98]. In addition, a study employing a mouse glioma model demonstrated that a mixture of four *Bifidobacterium* species impeded tumor growth by suppressing the MEK/ERK signaling pathway [99]. A brief schematic description of the main mechanisms of influence of microorganisms on tumors is presented in Figure 1.

## 4. The Relationship Between Immune Cells and Intratumoral Microbiota in the Context of Treatment Outcomes

The immunosuppressive TME compromises antineoplastic efficacy by combining physical and metabolic barriers (dense stroma, hypoxia, and CD38-driven adenosine accumulation) with disrupted antigen presentation and impaired T/NK-cell trafficking, producing poorly infiltrated tumors. In parallel, suppressive cell networks (regulatory T cells, MDSCs, and tumor-associated myeloid populations) and inhibitory checkpoint signaling (PD-1/CTLA-4) drive T-cell dysfunction/exhaustion, blunting cytokine production and cytotoxicity and thereby limiting responses to checkpoint blockade, adoptive cell therapies, chemotherapy, and immune-sensitizing combinations [100]. The intratumoral microbiota exerts a multifaceted influence on immune cells, shaping the unique immune microenvironment of tumors. Early in their interactions, microorganisms are recognized by the innate immune system through highly conserved pathogen-associated molecular patterns (PAMPs), including lipopolysaccharides (LPS), unmethylated double-stranded DNA, lipoproteins, single-stranded RNA, and flagellin [101]. These patterns are detected by pattern recognition receptors (PRRs), such as Toll-like receptors (TLRs) and Nod-like receptors (NLRs), initiating signaling cascades that regulate inflammation, activation of antigen-presenting cells, and subsequent immune response [102]. Transcriptional and immune profiles of tumor tissues demonstrate that a high load of intratumoral microbiota can significantly alter the composition of immune infiltration. Specifically, in nasopharyngeal carcinoma, an increased bacterial load is associated with decreased CD8^+^ T-cell infiltration, leading to a pronounced immunosuppressive effect and reduced antitumor activity of cytotoxic lymphocytes [103]. These effects are related to the fact that microbes are capable of not only causing inflammation, but also selectively suppressing key components of the adaptive immune system. Some bacteria use specific molecular mechanisms to evade immune surveillance. For example, *Helicobacter pylori* produce vacuolating cytotoxin A (VacA), which enhances cancer cell colonization while simultaneously suppressing T-cell proliferation. VacA disrupts antigen presentation by B cells, alters macrophage signaling pathways, and thereby weakens the immune system’s ability to destroy infected cells [104]. Similar immunosuppressive properties have been identified in *F. nucleatum*. This microorganism can interact with inhibitory receptors, including TIGIT (T cell immunoreceptor with Ig and ITIM domains) and CEACAM1 (carcinoembryonic antigen-related cell adhesion molecule 1), blocking natural killer (NK) cell cytotoxicity and T cell activity. This creates a microenvironment conducive to colorectal cancer progression [105]. Furthermore, in oral squamous cell carcinoma (OSCC), *F. nucleatum* activates the GalNAc-autophagy-TBC1D5 signaling pathway, which causes membrane GLUT1 accumulation, increases lactate production, and stimulates the formation of tumor-associated macrophages (TAMs). These macrophages enhance inflammatory metabolism and accelerate tumor growth [106].

Microbial metabolites also play a key role in regulating immune cells. In the tumor microenvironment, short-chain fatty acids, nucleosides, bile acid derivatives, and tryptophan catabolites influence dendritic cell maturation, T-cell polarization, and NK cell activity through receptor and epigenetic mechanisms [107]. These metabolites can both enhance the immune response and maintain immunosuppression depending on their concentration, local bioavailability, and interaction with immune cell receptors. In addition to their metabolic effects, bacteria can directly integrate into antigen-presentation mechanisms. Tumor cells are capable of presenting intracellular bacterial peptides on HLA class I and II molecules, creating additional antigenic targets for T cells [108]. This additional layer of antigenicity alters the specificity of T-cell receptors and their priming dynamics, complementing the action of tumor neoantigens [109]. Certain members of the oral microbiota, such as *Porphyromonas gingivalis*, possess pronounced immunomodulatory properties. Within tumor tissue, *P. gingivalis* can activate the NLRP3 inflammasome, inducing neutrophils to release elastase and promoting the formation of an inflammatory and immunosuppressive microenvironment [110]. Furthermore, stimulation of macrophages with *P. gingivalis* or its lipopolysaccharides is accompanied by a significant increase in the secretion of IL-1α, CCL3, and CCL5 [111]. Different bacterial strains induce different cytokine responses, indicating strain-specific immunomodulation. Single-nucleus RNA-sequencing (scRNA-seq) data reveal significant B cell heterogeneity in tumors and indicate that the intratumoral microbiome regulates their differentiation through the activation of genes including MMEL1 and CITED4 [112]. *Actinomyces* co-localizes with colorectal cancer-associated fibroblasts and reduces CD8^+^ T-lymphocyte infiltration into the tumor microenvironment by activating the TLR2/NF-κB pathway, thereby promoting tumor progression [113].

The study demonstrated that intratumoral microbiota shape tumor–immune interactions across multiple cancer types. In MMTV-PyMT mice, imaging revealed spatial segregation between microbes and CD4^+^/CD8^+^ T cells, indicating exclusion of activated T cells from bacterially colonized regions [28]. In the mice PDAC model, fungal components drove IL-33 upregulation and TH2/ILC2 recruitment, while IL-33 ablation or antifungal therapy reversed tumor growth [114]. In early-stage triple-negative breast cancer, higher microbial diversity and load correlated with increased CD4^+^CXCL13^+^ T cells, reduced tumor-associated macrophages, and better chemo-immunotherapy response [115]. In prostate cancer, *Cutibacterium* acnes was linked to Treg infiltration and macrophage PD-L1, CCL17, and CCL18 upregulation, supporting an immunosuppressive microenvironment that may promote tumor progression [116].

These results suggest the benefits of microbial interventions in the context of immunotherapy. However, further studies are needed in patients with different types of cancer. A summary of recent research findings on the relationship between the intratumoral microbiome and the immune response is presented in Table 1.

## 5. Current Limitation and Future Direction

Despite mounting evidence that the intratumoral microbiome plays a pivotal role in cancer progression and therapeutic response, several critical limitations impede the translation of these findings into clinical practice. A significant deficit in the extant literature pertains to the paucity of research addressing the influence of immune-cell therapies, including adoptive T-cell transfer, CAR-T or TIL therapies, on outcomes across diverse cancer types within the context of intratumoral microbial communities. To date, the majority of studies have been of an observational nature, having been limited to particular tumor types or immune checkpoint inhibitors, as opposed to cell therapies. For example, while the intratumoral microbiome has been implicated in shaping responses to immunotherapy, data specific to immune cell therapy remains scarce [124].

Mechanistically, the pathways by which intratumoral microbes modulate immune responses are only beginning to be elucidated. The complexity of microbial–immune–tumor cross-talk remains under-characterized, particularly in the setting of treated tumors, immune cell infusion, or following therapy-induced modifications of the tumor microenvironment. Although studies report involvement of pathways such as STING, TLR/NF-κB, β-catenin, and ROS signaling in microbe-driven immune modulation [125]. Furthermore, the relationship between extratumoral microbiomes (gut, oral cavity, and respiratory tract) and the intratumoral microbiome remains to be elucidated. The sources of the tumor-resident microbiome—whether via translocation, circulation, lymphatic spread, or local colonization—vary across tumor types and clinical contexts. In addition, the impact of these sources on intratumoral immune modulation remains unclear [97].

These limitations indicate several key avenues for future research. In order to elucidate the manner in which specific microbial taxa or their metabolites modulate the behavior of transplanted immune cells, including their persistence, exhaustion, phenotype, trafficking, and interactions with the host immune microenvironment, mechanistic studies employing advanced experimental systems are required. Future mechanistic studies should evaluate whether targeted depletion of protumor intratumoral consortia, such as *Fusobacterium*-rich biofilms in colorectal cancer or *Veillonella*-dominated communities in lung cancer, decreases CAR-T and TIL exhaustion marker expression (PD-1, TIM-3, LAG-3) and enhances immune cell persistence in tumor organoid co-culture and humanized mouse models [126]. Prospective clinical trials incorporating longitudinal microbiome sampling (tumor-intrinsic, gut, and oral) in patients receiving immune cell therapy are essential to delineate associations between microbial community states and therapeutic outcomes (response, persistence, and toxicity) and to evaluate the efficacy of microbiome-modulating interventions [127]. Comprehensive mapping of the interrelationships among systemic microbiomes (gut, oral, and skin), the intratumoral microbiome, and the immune landscape is necessary to identify upstream microbial drivers and causal pathways underlying immune modulation. The validation of microbial signatures as biomarkers or companion diagnostics for immune-based therapies, including immune cell therapies, should be prioritized, with an emphasis on assessing their additive predictive value beyond existing genomic, immunological and clinical parameters [128]. Prospective cellular immunotherapy trials should integrate intratumoral and systemic microbiome signatures with tumor-intrinsic (TMB, HLA genotype, PD-L1) and circulating biomarkers (ctDNA, immune cell clonality) to build multi-modal predictive models that stratify patients by likelihood of response [129,130]. Translational progress in this field will depend on the standardization of microbiome sampling and sequencing protocols, stringent contamination control, and the establishment of open data-sharing frameworks to enhance reproducibility and accelerate biomarker discovery [131]. Therapeutic innovation should focus on the rational development of microbiome-targeted adjuncts to immune cell therapy, such as engineered bacterial strains, bacteriophage-based strategies, or microbial metabolite modulators, while ensuring rigorous evaluation of their safety, off-target effects, and regulatory compliance. Clinical translation of engineered microbial therapeutics is already underway, with early-phase trials of SYNB1891 (an intratumoral *E. coli* Nissle strain producing a STING agonist under hypoxic conditions) demonstrating pharmacodynamic evidence of type I interferon activation and manageable safety in advanced solid tumors [132]. Complementary platforms—including probiotic-guided CAR-T systems where tumor-colonizing bacteria release synthetic CAR targets and chemokines, killed multi-receptor agonist bacterial products, and nanobody-decorated biohybrid bacteria—illustrate diverse avenues to harness microbial immunomodulation without sustained live colonization [133].

## 6. Conclusions

The intratumoral microbiome is a critical and functionally active component of the tumor microenvironment that significantly impacts cancer development, progression, and response to therapy. Microorganisms can either promote tumorigenesis through mechanisms like inducing genomic instability and fostering an immunosuppressive state or inhibit it by enhancing anti-tumor immunity. Future research should focus on mechanistic studies using multi-omics approaches, well-designed clinical trials to validate microbial biomarkers, and the rational development of microbiome-targeted interventions. Targeted modulation of the intratumoral and systemic microbiome may in the future form the basis for personalized combination strategies capable of overcoming resistance, enhancing treatment efficacy, and reducing adverse events.

## Figures and Tables

**Figure 1 cancers-18-00100-f001:**
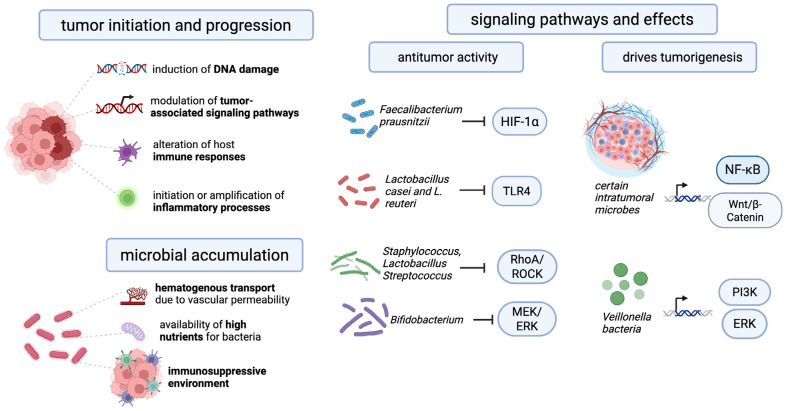
The key mechanisms and signaling pathways that explain the influence of the intratumoral microbiome on cancer development.

**Table 1 cancers-18-00100-t001:** Examples of studies aimed at studying the influence of intratumoral microorganisms and the immune response.

Year	Disease/ Model	Population	Brief Results	Implication for Therapy	References
2024	Colorectal cancer	Data from The Cancer Genome Atlas (TCGA) database	Network analysis revealed significant interactions between microbial abundance and genes involved in CTL evasion. Among these, suppressor of cytokine signaling 1 (SOCS1) exhibited the highest number of negative correlations, particularly with the genera *Phascolarctobacterium*, *Sneathia*, and *Intestinimonas*. Additionally, the genus *Oscillibacter* was negatively associated with exon skipping in the CD74 gene, indicating that the tumor-associated microbiota may influence the regulation of antigen presentation and thereby modulate the antitumor immune response. Furthermore, the analysis revealed that *Clostridium* was enriched in CRC patients who demonstrated resistance to ICB therapy.	The results suggest that profiling intratumor microbes alongside key tumor-infiltrating immune cells (notably MAIT cells) could be used in practice to stratify colorectal cancer patients by prognosis and likely responsiveness to immune checkpoint blockade.	Liu et al. [117]
2024	Breast cancer	Female MMTV-PyMT transgenic mice	Both three-dimensional imaging and X-Y optical sections revealed spatial segregation between the intratumor microbiome and CD4^+^ and/or CD8^+^ T cell clusters, indicating the exclusion of activated T cells from bacterially colonized tumor regions. These data are consistent with the observed isolation of TLS and NK cells from microbe-enriched areas, highlighting the spatial compartmentalization of immune and microbial niches within tumor tissue.	miCDaL enables centimeter-scale 3D mapping of the entire tumor and quantification of endemic intratumoral bacteria in both mouse and human tumors, eliminating a key practical bottleneck. This facilitates more reliable detection of microbiota-associated tumor niches and may aid in stratifying advanced disease stages or selecting microbiota-targeted therapies.	Wang et al. [28]
2022	Lung cancer	12 patients with early-stage lung cancer	The bacterial load was significantly higher in tumor cells compared to T cells, macrophages, other immune cells, and stromal components, forming a gradient that increased from normal lung tissue and tertiary lymphoid structures to tumor cells and the airways. This pattern suggests potential penetration of intratumoral bacteria through the respiratory tract. Moreover, bacterial load levels showed a positive correlation with the expression of oncogenic β-catenin, tumor histological type, and environmental exposures.	Bacterial load is highest in tumor cells (compared to immune/stromal cells) and closely correlates with tumor oncogenic pathways (in particular, with β-catenin), which supports the feasibility of therapeutic reduction in local intratumor bacterial load.	Wong-Rolle et al. [77]
2024	Colorectal cancer	C57BL/6 (B6) and Balb/c mice	The immunogenic chemotherapeutic agent oxaliplatin synergizes with *E. coli*, activating the innate and adaptive immune response in the colorectal tumor microenvironment, leading to complete remission and the formation of stable antitumor immunological memory in mice. The combined action of oxaliplatin and bacteria significantly enhances the expression of costimulatory and antigen-presenting molecules on antigen-presenting cells, facilitating the effective activation of cytotoxic T lymphocytes against tumor cells.	The combination of intratumoral *E. coli* with the immunogenic chemotherapeutic drug oxaliplatin (but not non-immunogenic 5-fluorouracil) can lead to complete tumor remission and induce durable antitumor immune memory associated with enhanced costimulation of antigen-presenting cells/antigen presentation and stronger CD8 T-cell activity.	Lim et al. [118]
2022	Colorectal cancer	C57BL/6 (B6) Thy 1.1 mice	The live attenuated *Brucella melitensis* strain (BmΔvjbR) was found to selectively colonize tumor tissue and remodel the tumor microenvironment by inducing proinflammatory polarization of M1 macrophages and enhancing both the number and activity of CD8^+^ cytotoxic T cells. In a colorectal adenocarcinoma model, treatment combining BmΔvjbR with adoptive transfer of tumor-specific CD8^+^ T cells almost completely suppressed tumor growth and achieved 100% animal survival. These findings highlight the potential of live attenuated bacteria to overcome tumor resistance to CAR-T therapy by remodeling the tumor microenvironment and activating macrophage-T-cell antitumor immunity.	An attenuated strain of BmvjbR migrates to tumors and shifts the tumor microenvironment toward proinflammatory immunity, and that the combination of BmvjbR with specific anti-CEA CAR-T cells in a mouse model of colon cancer almost completely suppresses tumor growth, with 100% survival reported.	Guo et al. [119]
2023	Leukemia	Non-obese diabetic scid gamma mice	A probiotic-targeted CAR-T cell (ProCAR) platform was developed in which tumor-colonizing probiotics secrete synthetic targets that mark tumor tissue for local lysis by CAR-T cells. Using the *Escherichia coli* Nissle 1917 strain with a synchronized lysis system (SLIC) enabled the release of synthetic targets directly into the tumor microenvironment, inducing safe and effective CAR-T cell activation in various cancer models. Additionally, an engineered strain co-expressing the chemokine mutant CXCL16^K42A^ enhanced ProCAR-T cell recruitment and antitumor activity, resulting in increased hCD45^+^CD3^+^ T cell infiltration and significant tumor growth inhibition without toxic effects.	ProCAR, in which tumor-colonizing probiotic *E. coli* bacteria release synthetic CAR targets (and, in improved strains, chemokines) within tumors, enables antigen-independent activation/lysis of CAR T cells. Safety and antitumor efficacy have been demonstrated in various xenograft and syngeneic models of solid tumors.	Vincent et al. [120]
2021	Melanoma	C57BL/6NTac germ-free, BALB/cAnNCrl, B6-Ly5.1/Cr, B6-Ifnar1 (Ifnar1 KO) and C57BL/6J-Tmem173/J (STING KO) mice, 6 patients with melanoma	The microbiota regulates the immune compartment of the tumor microenvironment, reprogramming mononuclear phagocytes into immune-stimulatory monocytes and dendritic cells. The absence of microbiota shifts the balance of the tumor microbiome toward pro-tumorigenic macrophages, while microbial STING agonists induce type I interferon production, regulating macrophage polarization and NK cell–dendritic cell interactions. Modulation of the microbiota with a high-fiber diet activated the IFN-I–NK–DC axis and enhanced the efficacy of immune checkpoint blockade therapy, as confirmed in both experimental models and patients with melanoma.	Microbiota-derived STING agonists (particularly cyclic dinucleotides such as c-di-AMP, including those from Akkermansia muciniphila) stimulate intratumoral production of type I interferon by monocytes, which shifts the tumor microenvironment from pro-tumor macrophages to immune-stimulatory monocytes/dendritic cells and activates the NK–DC axis, improving the response to therapies such as immune checkpoint blockade.	Lam et al. [121]
2022	Pancreatic cancer	C57BL/6 mice	Oncogenic Kras^G12D^ was shown to induce IL-33 expression in pancreatic ductal adenocarcinoma cells, which promotes the recruitment and activation of TH2 and ILC2 cells, which stimulate tumor growth. Ablation of IL-33 in tumor cells or antifungal therapy reduced TH2 and ILC2 cell infiltration, induced tumor regression, and increased survival. Thus, the intratumor mycobiome regulates IL-33 secretion and promotes the formation of a protumorigenic environment, opening up opportunities for targeted therapy for PDAC.	IL-33–ST2/type-2-immune axis and/or modulating tumor-associated fungi could reduce tumor burden and extend survival (shown preclinically), and that tumor IL-33 may help stratify patients for such interventions.	Alam et al. [114]
2025	Breast cancer	89 female patients	In patients with early-stage TNBC treated with neoadjuvant chemo-immunotherapy, the pCR group exhibited higher intratumoral microbiota diversity and load compared to the non-pCR group. Single-cell RNA sequencing revealed enhanced T cell infiltration and reduced tumor-associated macrophages in the pCR group. Microbiota load positively correlated with CD4^+^CXCL13^+^ T cells and negatively with CD68^+^SPP1^+^ macrophages. Combined 16S and scRNA-seq analyses confirmed bacterial presence in both cancer and immune cells. A predictive model integrating microbial and clinicopathological data accurately forecasted pCR outcomes.	Higher intratumor microbiota/diversity is associated with achieving pathological complete response (pCR) and with a more immune-activated tumor microenvironment (more T-cell infiltration, fewer tumor-associated macrophages).	Chen et al. [115]
2022	Cutaneous melanoma	Data from The Cancer Genome Atlas (TCGA) database	Low CD8^+^ T cell counts were associated with worse patient survival (OR = 1.57; 95% CI: 1.17–2.10; *p* = 0.002). The *Lachnoclostridium* genus showed the highest positive correlation with CD8^+^ T cell infiltration and expression of chemokines CXCL9, CXCL10, and CCL5, and its high level was associated with a reduced risk of mortality (*p* = 0.0003).	Both higher CD8 T cell infiltration and higher intratumoral *Lachnoclostridium* abundance were associated with better overall survival, supporting the idea that modulation of the intratumoral/gut microbiome could be used to “activate” tumors and potentially improve immunotherapy outcomes.	Zhu et al. [122]
2021	Prostate cancer	137 male patients	Macrophages stimulated with *Cutibacterium acnes* in vitro were shown to increase the expression of PD-L1, CCL17, and CCL18 (*p* < 0.05), and the presence of *C. acnes* in prostate cancer patients was positively correlated with Tregs infiltration in the tumor stroma and epithelium (*p* = 0.0004 and *p* = 0.046). These data suggest that *C. acnes* contribute to the formation of an immunosuppressive tumor microenvironment that promotes prostate cancer progression.	*C. acnes* may contribute to the establishment or maintenance of an immunosuppressive prostate tumor microenvironment (via macrophage polarization/chemokine signaling and Treg recruitment), suggesting a potential microbial target or stratification factor for future prostate cancer immunomodulation strategies.	Davidsson et al. [116]
2018	Pancreatic cancer	KC, C57BL/6 (H-2Kb) mice	Pancreatic tumor tissue in mice and humans contains a significantly more abundant microbiome compared to normal tissue, with certain bacterial taxa selectively enriched compared to the intestine. Microbiome ablation prevented the development of pancreatic ductal adenocarcinoma and induced immune reprogramming of the microenvironment with increased M1 macrophage polarization and activation of CD4^+^ Th1 and CD8^+^ T cells. Furthermore, microbiome ablation enhanced the efficacy of immunotherapy by upregulating PD-1 expression, while the PDA microbiome induced a tolerogenic phenotype through activation of specific Toll-like receptors.	Microbiome ablation can enhance the sensitivity of PDAC to PD-1 checkpoint inhibitor therapy (a synergistic effect compared to either modality alone), supporting the feasibility of targeted microbiome modulation as a practical adjunctive strategy to overcome immunosuppression in PDAC.	Pushalkar et al. [58]
2019	Pancreatic cancer	68 patients	Long-term survivor patients with pancreatic ductal adenocarcinoma exhibited higher alpha diversity of the tumor microbiome and a characteristic microbial signature (*Pseudoxanthomonas*–*Streptomyces*–*Saccharopolyspora*–*Bacillus clausii*) predicting a favorable prognosis. Transplantation of microbiota from long-term survivor donors into mice slowed tumor growth and enhanced immune infiltration. Immunohistochemistry analysis revealed that long-term survivor patients had significantly higher densities of CD3^+^, CD8^+^, and GzmB^+^ T cells (*p* = 0.0273; *p* < 0.0001; *p* = 0.04), which positively correlated with both overall survival and microbiome diversity, suggesting a link between microbial diversity, CD8^+^ T cell activation, and the antitumor immune response.	Fecal microbiota transplantation from long-term survivors can alter the tumor microbiome, enhance CD8 T cell-mediated antitumor immunity, and reduce tumor growth in mouse models, suggesting microbiome modulation as a potential adjunctive therapeutic strategy for pancreatic cancer.	Riquelme et al. [123]

## Data Availability

No new data were created or analyzed in this study.

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
