# Peer review of "Intratumoral Microbiome: Impact on Cancer Progression and Cellular Immunotherapy"

_cancers, 2025, doi:10.3390/cancers18010100_

Round 1

Reviewer 1 Report

Comments and Suggestions for Authors

refer attachment
Intratumoral Microbiome: Impact on Cancer Progression and Cellular Immunotherapy
Overview
This review examined how the intratumoral microbiome—comprising bacteria, fungi, and
viruses—can influence cancer progression by inducing DNA damage, modulating oncogenic
signaling pathways, and shaping anti-tumor immune responses. Further, it highlighted the
microbiome’s complex role in cellular immunotherapies and calls for future research to focus on
mechanistic studies and microbiome-targeted interventions to improve therapeutic outcomes.
Major Comments
1. The review established the foundational concept that the intratumoral microbiome is a real
and active component of the TME with demonstrable effects on cancer biology. The review
would be strengthened by a more critical discussion of the technical challenges (e.g., low
biomass, contamination).
2. Authors did an excellent job of cataloging the various mechanisms (e.g., DNA damage,
signaling pathway modulation) by which intratumoral microbes can influence cancer
progression. However, the volume of mechanisms and examples are overwhelming. A
clearer hierarchical framework distinguishing primary from secondary mechanisms would
greatly enhance clarity for the reader.
3. The review highlighted the bidirectional role of the microbiome, detailing both pro-
tumorigenic and anti-tumorigenic functions of different microbes. However, the discussion
presented these effects as a list. A more in-depth analysis of the contextual factors (e.g.,
cancer type, host genetics, microbial community structure) that determine whether a
microbe exerts a net positive or negative effect would be valuable.
4. The inclusion of a dedicated section on the origin of the intratumoral microbiota and its
connection to other body sites (e.g., gut-tumor axis) is a major strength, providing a systemic
view. This section could be more precise by mapping the proposed routes of translocation
(hematogenous, direct invasion, etc.) to specific cancer types and citing the evidence for
each pathway.

5. Table 1 is insightful that provided evidence linking specific microbes to immune responses.
However, the table’s formatting is inconsistent and difficult to follow. Standardizing the
columns (e.g., always including ‘key finding’ and ‘implication for therapy’) and ensuring
all details are complete would drastically improve its purpose.
6. Figure 1 is a commendable effort to visually summarize the complex signaling pathways
discussed in the text. The figure as it is currently very dense and text-heavy, reducing its
visual impact. A redesign using clearer icons for cells and pathways, and better spatial
organization to show crosstalk, would make it more effective.
7. The ‘current limitations and future direction’ section correctly identified key gaps, such as
the need for mechanistic studies and clinical trials. The future directions should be more
forward-thinking and specific. For instance, it could propose testable hypotheses, discuss
the potential and challenges of engineered microbial therapeutics, or explore how
microbiome data could be integrated with other biomarkers in clinical decision-making.

Minor Comments
1. The last sentence of the abstract is general. Tuning it to more specifically mention ‘cellular
immunotherapy’ would mirror the title and strengthen the abstract’s focus.
2. The label ‘accumulation’ in the top box (Figure 1) is a bit vague.
3. The conclusion could be strengthened by adding a forward-looking phrase about the
potential of harnessing the microbiome, making it more future oriented.

Remark
The review is well written and presented. Including the above stated suggestions will improve the manuscript further. Please check for minor typo and grammatical errors.

Author Response

The authors very much appreciated the constructive comments on this manuscript by the reviewer. The comments have been very thorough and useful in improving the manuscript.

Comment 1. The review established the foundational concept that the intratumoral microbiome is a real
and active component of the TME with demonstrable effects on cancer biology. The review
would be strengthened by a more critical discussion of the technical challenges (e.g., low
biomass, contamination).

Response 1. The authors agree with this comment, and the relevant information has been added to the Introduction section (line 118-124).

Comment 2.  Authors did an excellent job of cataloging the various mechanisms (e.g., DNA damage,
signaling pathway modulation) by which intratumoral microbes can influence cancer
progression. However, the volume of mechanisms and examples are overwhelming. A
clearer hierarchical framework distinguishing primary from secondary mechanisms would
greatly enhance clarity for the reader.

Response 2. We've added subsections to the section: The Role of the Intratumoral Microbiome in Cancer Progression. We've also added information on primary and secondary mechanisms and made a number of corrections and additions.

Comment 3.  The review highlighted the bidirectional role of the microbiome, detailing both pro-
tumorigenic and anti-tumorigenic functions of different microbes. However, the discussion
presented these effects as a list. A more in-depth analysis of the contextual factors (e.g.,
cancer type, host genetics, microbial community structure) that determine whether a
microbe exerts a net positive or negative effect would be valuable.

Response 3. The authors agree with this comment, and we have made appropriate changes to the text of the manuscript.

Comment 4.  The inclusion of a dedicated section on the origin of the intratumoral microbiota and its
connection to other body sites (e.g., gut-tumor axis) is a major strength, providing a systemic
view. This section could be more precise by mapping the proposed routes of translocation
(hematogenous, direct invasion, etc.) to specific cancer types and citing the evidence for
each pathway.

Response 4. The authors agree with this comment, and we have made appropriate changes to the text of the manuscript.

Comment 5. Table 1 is insightful that provided evidence linking specific microbes to immune responses.
However, the table’s formatting is inconsistent and difficult to follow. Standardizing the
columns (e.g., always including ‘key finding’ and ‘implication for therapy’) and ensuring
all details are complete would drastically improve its purpose.

Response 5. We have added a column "Implication for therapy" to the table.

Comment 6. Figure 1 is a commendable effort to visually summarize the complex signaling pathways
discussed in the text. The figure as it is currently very dense and text-heavy, reducing its
visual impact. A redesign using clearer icons for cells and pathways, and better spatial
organization to show crosstalk, would make it more effective.

Response 6. We agree with this comment, Figure 1 has been corrected.

Comment 7.  The ‘current limitations and future direction’ section correctly identified key gaps, such as
the need for mechanistic studies and clinical trials. The future directions should be more
forward-thinking and specific. For instance, it could propose testable hypotheses, discuss
the potential and challenges of engineered microbial therapeutics, or explore how
microbiome data could be integrated with other biomarkers in clinical decision-making.

Response 7. The authors agree with this comment, and we have made appropriate changes to the text of the manuscript.

Comment 8. The last sentence of the abstract is general. Tuning it to more specifically mention ‘cellular
immunotherapy’ would mirror the title and strengthen the abstract’s focus.

Response 8. The authors agree with this comment, and we have made appropriate changes to the text of the manuscript.

Comment 9. The label ‘accumulation’ in the top box (Figure 1) is a bit vague

Response 9. Corrected. 

Comment 10.  The conclusion could be strengthened by adding a forward-looking phrase about the
potential of harnessing the microbiome, making it more future oriented.

Response 10. The authors agree with this comment, and we have made appropriate changes to the text of the manuscript.

Reviewer 2 Report

Comments and Suggestions for Authors

Overall, the this review article is giving comprehensive data on interaction between microbiota and carsinogenesis, tumor progression and invasion. The focus is in bacteria. The figure and table are highly wellcomed and allows to shorten the text body. The manuscript is missing a chapter ’Material and Methods’ to describe the methodology to create this manuscript. Furthermore, there is no scrutinized discussion of the methodology, its benefits and limitations with proposal for any improvement when writing a review article. This addition will improve the scientific value of the manuscript, and avoid ’driven’ or ’false’ conclusions. The chapter 5 is proposed to get a revised titlem ’Discussion’ and to include the authors’ evaluation of the used methodology.

Major comments

  1. The manuscript is missing a chapter ’Material and Methods’. The following topics are expected to be added:
  2. Which data sources where used? How were they choosen? Was gov database used?
  3. Which keywords were used for the search of publications, documents, etc in these databanks? What was the basis to choose the keywords? Please, complement the list of keywords if additional ones were used than listed on page 1 of the manuscript.
  4. Did the authors perform a pilot study using lower and/or higher number of keywords before ending up with those listed for this review article? If a pilot study was performed, please, add this information and a short discussion (chapter ’Discussion’) why the final decision included the listed keywords.
  5. Did each author study all the used references or whether the reading performed in another way?
  6. Was there need for any consensus discussion due to conflicts in interpretation of publications and their various conclusions on the same topic? What was the predefined method to come into a consensus in these situations? Were any independent expert involved in these situations in need for a consensus?
  7. Was the quality of included literature formally evaluated? There are at least two common methods to perform search for a review article: i) journal method search based on the listed keywords, and ii) reference list search method. Please, define if one of them or another was implemented. Please, discuss (chapter ’Discussion’) benefits if any of the journal method search as compared to so called reference list search method? If both methods were combined then obviously one of the methods has limitations the authors wanted to cover by adding the reference list method (usually by screening of the reference lists of the choosen publications). The quality and comprehensiveness of publication lists for the journal based method influence obviously extensively the outcome.
  8. Did the authors check the used histological, immunohistological and genomic analyse methods in the choosen publications? If different methods and different quality of ’a same method’ for the level of sensitivity and specificity were identified, the findings will influence the conclusions in the very specific publications and further the authors’ summaries of findings in these used publications as well as the ultimate conclusion. Please, expand the information in the chapter ’Material and Methods’ and add your evaluation in the chapter ’Discussion’.

  1. The manuscript is focusing on especially bacteria. Please, present this limitation already in the chapters ’Short summary’ and ’Introduction’. The writing of the abstract should give the limitation with clear and precise writing statements where nothing about virus and fungi will be mentioned, making the focus in bacteria clear.

  1. Some of cancers are know to have, especially, a virus as trigger for the tumorgenesis. Since this isn’t specifically mentioned and authors seem not want to include this aspect in this review as understood what is especially presented in the third subsection of chapter 3, the authors are advised to shortly mention this in the chapter, Introduction, and give a statement that it is outside the scope of the present manuscript (please, see also my major comment nr 1).

  1. Please, revise the expression of ’dendritic cell vaccine’ (eg. chapter ’Introduction’, line 53).

Vaccines are defined to be used as prophylaxis against infectious diseases , i.e. to prime or boost an immune response against an infectious agent (Guideline on clinical evaluation of new vaccines). The authors have initially used the perfectly right term: ’immunotherapy... as treatment strategy…’. Thus, dendritic cell immunotherapy is to be used when used for treatment of malignances. Furthermore, in this context it’s adviced also to specify that cells (autologous or allogeneic) used in their natural form without any manipulation is defined cell therapy as transplantation of cells via infusion, in situ application, etc (line 55).

  1. The authors are requested to use correct regulatory terminology to improve readers’ knowledge to understand the difference between cell and gene therapy medicinal products versus transplantation of cells, organs and tissues. Both of these can be used for various indications, i.e. as therapy. The U.S. Food and Drug Administration (FDA) and the European Medicines Agency (EMA) doesn’t approve therapies but medicinal products. Please, revise for example on lines 62-63 to write ”… several cell-based medicinal products…”, and check the remaining text body for this topic. Here and there it writes for example ’transplanted immune cells’ but actually both transplanted cells and cell therapy medicinal products are to be included.

  1. Connected to the previous major comments nr. 4 and 5, the authors may consider to add a fact box where central terminology used in the manuscript is defined based on the regulatory and scientific basis; for example gene and cell therapy medicinal product, transplantation, immunotherapy, immune cell therapy, vaccine, regulatory approval = marketing authorisation by…, carsinogenesis, etc.

  1. The authors are advised to present the most important safety findings of the cell therapy medicinal products’ use instead only to give litterature references 14-17 (lines 72-74).

  1. The presentation of mechanisms of microbes' entry into the tumor in the first subsection of the chapter 3, is loosing the logical focus in the sentence: ’Immune evasion by cancer cells creates an immunosuppressive environment… (lines 198-200). This sentence is awaited to end by referring what is happening with microorganisms, not about immune evasion of cancer cells. Please, revise.

  1. The statement of the predominant existence of bacteria inside of cells is expected to follow additional information about the input of this phenomena or at least a reference to later sectionto be added in the text body of the manuscript where this is discussed (lines 196-197). Are bacteria located mostly in cancer cells or in cells of the supporting tissue? Is the phenomena stabel or transient? Historically few bacteria are defined to be intracellular, eg. Mucobacterium t

  1. The presented results in various publications where divergent outcomes were identified in preclinical studies and clinical trials is expected to be mentioned when presenting these specific situations and futher discussed (chapter ’DIscussion’). What is the authors’ opinion on the findings (chapters 2-4) , and proposals for future scientific approach in these situations?

Minor comments

  1. Please, check the use of abbreviations. In general, an abbreviation, if used, is given after a word when it’s used for the first time. Late the abbreviation is used, not the word, but in the beginning of a sentence where usually an abbreviation isn’t used.

Thus, for example ’Non-specific OCI…’ (line 56) needs to be complemented. Please, check the whole text body for this subject.

  1. The writing in the chapter 5 might benefit by revision of the title to ’Discussion’ and to include above required additional topics to be discussed in subsections.

  1. Is the following sentence missing some words? I’ve added a proposal. Please, check if my understanding of your purpose is right.

’Clinical data underscores the deleterious effect of antibiotics administered given prior to cancer treatment on patients’ outcomes and survival. ’ (lines 88-89).

  1. The text body of the manuscript doesn’t refer to the Figure 1 and Table 1 which are collecting important data, and highly supported to be included. Please, add reference to the figure and table in approriate chapters.

Furthermore, please, consider to shorten the text in the respective chapters by reffering to especially the Tabel 1 to easy readers to identify the ’take-home data’. This is a comprehensive review and thus, would benefit to scrutinize which data is necessary to desripe in details versus when it’s enough to refer to litterature and to the figure and table.

  1. There is an extra werb in the following sentence: ’ In addition, the impact of these sources …’ (line 390). Please, delete either ’is’ or ’remains’.

Author Response

The authors very much appreciated the constructive comments on this manuscript by the reviewer. The comments have been very thorough and useful in improving the manuscript.

Comment 1. The manuscript is missing a chapter ’Material and Methods’. The following topics are expected to be added:  Which data sources where used? How were they choosen? Was gov database used?

Comment 2. Which keywords were used for the search of publications, documents, etc in these databanks? What was the basis to choose the keywords? Please, complement the list of keywords if additional ones were used than listed on page 1 of the manuscript.

Comment 3. Did the authors perform a pilot study using lower and/or higher number of keywords before ending up with those listed for this review article? If a pilot study was performed, please, add this information and a short discussion (chapter ’Discussion’) why the final decision included the listed keywords.

Comment 4. Did each author study all the used references or whether the reading performed in another way?

Comment 5. Was there need for any consensus discussion due to conflicts in interpretation of publications and their various conclusions on the same topic? What was the predefined method to come into a consensus in these situations? Were any independent expert involved in these situations in need for a consensus?

Comment 6. Was the quality of included literature formally evaluated? There are at least two common methods to perform search for a review article: i) journal method search based on the listed keywords, and ii) reference list search method. Please, define if one of them or another was implemented. Please, discuss (chapter ’Discussion’) benefits if any of the journal method search as compared to so called reference list search method? If both methods were combined then obviously one of the methods has limitations the authors wanted to cover by adding the reference list method (usually by screening of the reference lists of the choosen publications). The quality and comprehensiveness of publication lists for the journal based method influence obviously extensively the outcome.

Comment 7. Did the authors check the used histological, immunohistological and genomic analyse methods in the choosen publications? If different methods and different quality of ’a same method’ for the level of sensitivity and specificity were identified, the findings will influence the conclusions in the very specific publications and further the authors’ summaries of findings in these used publications as well as the ultimate conclusion. Please, expand the information in the chapter ’Material and Methods’ and add your evaluation in the chapter ’Discussion’.

 Response 1-7. We apologize for the unclear description of the article's objectives. This manuscript is a narrative review. Although we would be interested in conducting a systematic review/meta-analysis (using a specific methodology, such as PRISM) on this topic, we did not set such a goal for this manuscript. However, we believe it would be useful to include some information about the sources of the literature search.

Comment 8. The manuscript is focusing on especially bacteria. Please, present this limitation already in the chapters ’Short summary’ and ’Introduction’. The writing of the abstract should give the limitation with clear and precise writing statements where nothing about virus and fungi will be mentioned, making the focus in bacteria clear.

 Response 8.  The authors agree with this comment, and we have made appropriate changes to the text of the manuscript.

Comment 9. Some of cancers are know to have, especially, a virus as trigger for the tumorgenesis. Since this isn’t specifically mentioned and authors seem not want to include this aspect in this review as understood what is especially presented in the third subsection of chapter 3, the authors are advised to shortly mention this in the chapter, Introduction, and give a statement that it is outside the scope of the present manuscript (please, see also my major comment nr 1).

 Response 9. Yes, you're right. This review focuses more on the bacterial composition of the microbiome. We've added some information about the impact of viruses on cancer development in Section 3.

Comment 10. Please, revise the expression of ’dendritic cell vaccine’ (eg. chapter ’Introduction’, line 53).

Vaccines are defined to be used as prophylaxis against infectious diseases , i.e. to prime or boost an immune response against an infectious agent (Guideline on clinical evaluation of new vaccines). The authors have initially used the perfectly right term: ’immunotherapy... as treatment strategy…’. Thus, dendritic cell immunotherapy is to be used when used for treatment of malignances. Furthermore, in this context it’s adviced also to specify that cells (autologous or allogeneic) used in their natural form without any manipulation is defined cell therapy as transplantation of cells via infusion, in situ application, etc (line 55).

Response 10.  The authors agree with this comment, and we have made appropriate changes to the text of the manuscript. 

Comment 11. The authors are requested to use correct regulatory terminology to improve readers’ knowledge to understand the difference between cell and gene therapy medicinal products versus transplantation of cells, organs and tissues. Both of these can be used for various indications, i.e. as therapy. The U.S. Food and Drug Administration (FDA) and the European Medicines Agency (EMA) doesn’t approve therapies but medicinal products. Please, revise for example on lines 62-63 to write ”… several cell-based medicinal products…”, and check the remaining text body for this topic. Here and there it writes for example ’transplanted immune cells’ but actually both transplanted cells and cell therapy medicinal products are to be included.

Response 11.  The authors agree with this comment, and we have made appropriate changes to the text of the manuscript.

 Comment 12. Connected to the previous major comments nr. 4 and 5, the authors may consider to add a fact box where central terminology used in the manuscript is defined based on the regulatory and scientific basis; for example gene and cell therapy medicinal product, transplantation, immunotherapy, immune cell therapy, vaccine, regulatory approval = marketing authorisation by…, carsinogenesis, etc.

Response 12. We have made a number of corrections to terminology and definitions.

Comment 13. The authors are advised to present the most important safety findings of the cell therapy medicinal products’ use instead only to give litterature references 14-17 (lines 72-74).

Response 13. We have expanded the information on the principles for determining the biosafety of cell therapy in the Introduction section.

Comment 14. The presentation of mechanisms of microbes' entry into the tumor in the first subsection of the chapter 3, is loosing the logical focus in the sentence: ’Immune evasion by cancer cells creates an immunosuppressive environment… (lines 198-200). This sentence is awaited to end by referring what is happening with microorganisms, not about immune evasion of cancer cells. Please, revise.

Response 14. The authors agree with this comment, and we have made appropriate changes to the text of the manuscript.

Comment 15. The statement of the predominant existence of bacteria inside of cells is expected to follow additional information about the input of this phenomena or at least a reference to later sectionto be added in the text body of the manuscript where this is discussed (lines 196-197). Are bacteria located mostly in cancer cells or in cells of the supporting tissue? Is the phenomena stabel or transient? Historically few bacteria are defined to be intracellular, eg. Mucobacterium t

Response 15. The authors agree with this observation. We have expanded our information on the intracellular localization of microorganisms in the tumor microenvironment.

Comment 16. The presented results in various publications where divergent outcomes were identified in preclinical studies and clinical trials is expected to be mentioned when presenting these specific situations and futher discussed (chapter ’DIscussion’). What is the authors’ opinion on the findings (chapters 2-4) , and proposals for future scientific approach in these situations?

Response 16. While differences in results between studies are indeed intriguing, our goal was not to conduct a systematic review. We provided a brief summary of the experimental studies' findings and attempted to draw general conclusions.

 Comment 17. Please, check the use of abbreviations. In general, an abbreviation, if used, is given after a word when it’s used for the first time. Late the abbreviation is used, not the word, but in the beginning of a sentence where usually an abbreviation isn’t used.

Thus, for example ’Non-specific OCI…’ (line 56) needs to be complemented. Please, check the whole text body for this subject.

 Response 17. The authors agree with this comment, and we have made appropriate changes to the text of the manuscript.

Comment 18. The writing in the chapter 5 might benefit by revision of the title to ’Discussion’ and to include above required additional topics to be discussed in subsections.

 Response 18. While it would be useful to discuss the results of experimental articles, a narrative review cannot provide a single conclusion for comparison with other works.

Comment 19. Is the following sentence missing some words? I’ve added a proposal. Please, check if my understanding of your purpose is right.

’Clinical data underscores the deleterious effect of antibiotics administered given prior to cancer treatment on patients’ outcomes and survival. ’ (lines 88-89).

Response 19. Corrected

Comment 20. The text body of the manuscript doesn’t refer to the Figure 1 and Table 1 which are collecting important data, and highly supported to be included. Please, add reference to the figure and table in approriate chapters.

Furthermore, please, consider to shorten the text in the respective chapters by reffering to especially the Tabel 1 to easy readers to identify the ’take-home data’. This is a comprehensive review and thus, would benefit to scrutinize which data is necessary to desripe in details versus when it’s enough to refer to litterature and to the figure and table.

 Response 20. We have added a column "Implication for therapy" to the table. The text describes only a portion of the studies listed in the table. We also made a number of changes to the structure of the test.

Comment 21. There is an extra werb in the following sentence: ’ In addition, the impact of these sources …’ (line 390). Please, delete either ’is’ or ’remains’.

Response 21. Corrected

Reviewer 3 Report

Comments and Suggestions for Authors

The manuscript represent an interesting review on microbiome impact on cancer, cancer progression and immunotherapy. The rationale is adequate and the general description is updated. The only point that is missing is the enphasis on cancer resistance and therapy which would be very important for the reader, specially the non expert ones. The conclusions are sound.

Author Response

The authors are grateful for the high evaluation of our manuscript.

Comment 1. The manuscript represent an interesting review on microbiome impact on cancer, cancer progression and immunotherapy. The rationale is adequate and the general description is updated. The only point that is missing is the enphasis on cancer resistance and therapy which would be very important for the reader, specially the non expert ones. The conclusions are sound.

Response 1. We have added information on the influence of the immunosuppressive tumor microenvironment on the effectiveness of cancer therapy to the text of the manuscript.

Round 2

Reviewer 2 Report

Comments and Suggestions for Authors

Thank you for revising the manuscript based on reviewers’ comments and proposals. No additional comments from my perspective.